

# Spatial metabolomics shows contrasting phosphonolipid distributions in tissues of marine bivalves

Patric Bourceau[1,2], Dolma Michellod[1], Benedikt Geier[1] and Manuel Liebeke[1]

[1] Max Planck Institute for Marine Microbiology, Bremen, Germany
[2] MARUM—Center for Marine Environmental Sciences of the University of Bremen, Bremen, Germany

## ABSTRACT

Lipids are an integral part of cellular membranes that allow cells to alter stiffness, permeability, and curvature. Among the diversity of lipids, phosphonolipids uniquely contain a phosphonate bond between carbon and phosphorous. Despite this distinctive biochemical characteristic, few studies have explored the biological role of phosphonolipids, although a protective function has been inferred based on chemical and biological stability. We analyzed two species of marine mollusks, the blue mussel *Mytilus edulis* and pacific oyster *Crassostrea gigas*, and determined the diversity of phosphonolipids and their distribution in different organs. High-resolution spatial metabolomics revealed that the lipidome varies significantly between tissues within one organ. Despite their chemical similarity, we observed a high heterogeneity of phosphonolipid distributions that originated from minor structural differences. Some phosphonolipids are ubiquitously distributed, while others are present almost exclusively in the layer of ciliated epithelial cells. This distinct localization of certain phosphonolipids in tissues exposed to the environment could support the hypothesis of a protective function in mollusks. This study highlights that the tissue specific distribution of an individual metabolite can be a valuable tool for inferring its function and guiding functional analyses.

## INTRODUCTION

Lipids are a diverse class of biomolecules universally present across all kingdoms of life. They are building blocks of cellular membranes, store chemical energy, and are crucial effector molecules of the cell cycle, for instance by inducing proliferation and inhibition of apoptosis (*Hoeferlin, Wijesinghe & Chalfant, 2013*). The physio-chemical properties and the localization of a given lipid within a cell depend on the lipid's molecular structure, but a lipid's biological function cannot be predicted from chemical structure alone. Some lipids, like phosphatidylcholines (PC) and phosphatidylethanolamines (PE), are ubiquitous components in eukaryotic membranes, whereas other lipids are specific to individual species and their lifestyles (*Corcelli et al., 2004*; *Dembitsky & Levitsky, 2004*; *Van Meer, Voelker & Feigenson, 2008*). The diversity of lipids with common headgroups like

Corresponding author
Manuel Liebeke,
mliebeke@mpi-bremen.de

PCs and PEs is well represented in in current metabolite databases (*e.g.*, LIPIDMAPS (*Fahy et al., 2007*; *Sud et al., 2006*) (accessed November 23rd 2021).

In contrast, lipids with uncommon headgroups such as arseno-, phosphono- or sulfo-lipids have often been overlooked in standard lipidomics workflows. The major difficulty in analyzing these less common lipids stems from an underrepresentation of their chemical diversity in databases used for metabolite annotation.

Analyzing lipid inventories with liquid chromatography mass spectrometry (LC-MS) on extracted tissue samples is an established lipidomics approach (*Long et al., 2020*). While highly sensitive, LC-MS studies reveal averaged lipid profiles of homogenized samples, so differentiating cell- and tissue-specific metabolomes remains challenging. However, determining these location-specific metabolomes can be critical when defining the metabolic phenotype and revealing the function of individual cell types of an organ (*Guo, Zhang & Le, 2021*). Spatial metabolomics, enabled by techniques such as matrix-assisted laser desorption ionization mass spectrometry imaging (MALDI-MSI), can locate metabolites in tissues (*Rappez et al., 2021*) and complement our knowledge on the functions of lipids and other metabolites (*Ellis et al., 2017*; *Geier et al., 2020*; *Liebeke et al., 2015*). These technologies recently reached single cell resolution; it is now possible to link individual cell types to their characteristic lipidome (*Niehaus et al., 2019*; *Rappez et al., 2021*) and detect metabolic heterogeneity even within the same cell type (*Geier et al., 2020*; *Prade et al., 2020*).

While modern mass spectrometry (MS) methods can detect hundreds of signals from one sample, their annotation often follows an automated approach using a variety of tools (*Misra, 2021*) that rely on metabolite databases such as HMDB or LIPIDMAPS (*Fahy et al., 2007*; *Wishart et al., 2018*). Still, the majority of peaks remains unidentified in MS experiments, and has been called "dark metabolome" (*da Silva, Dorrestein & Quinn, 2015*). In addition to classic compound databases, combinatorial chemistry allows for the identification of new structures and exploit small modifications of known metabolites, such as the degree of saturation or the chain length of a fatty acid, to elucidate the dark metabolome through dereplication (*Wang et al., 2016*). Despite those improvements, entire known lipid classes are still underrepresented in common metabolite/lipid databases (*Aimo et al., 2015*; *Fahy et al., 2007*; *Sud et al., 2006*) and missed by database-dependent annotation workflows.

Among these overlooked lipids are phosphonolipids, a class of lipids characterized by a carbon-phosphorous bond, *i.e.,* a phosphonate moiety, in their polar headgroup. The phosphonate headgroup is linked to a typical fatty acid backbone. For all known phospholipids there is a potential phosphonolipid analog. For example, replacing the phosphoethanolamine (PE) or phosphocholine (PC) headgroup with phosphonoethanolamine or phosphonocholine would produce the corresponding phosphonolipids (PnE and PnC). In the same fashion, the phosphonolipid analogs of ceramides (PnE-Cer) combine a sphingolipid base with phosphonoethanolamine. PnE, PnC and PnE-Cer have been found across phyla, from bacteria to eukaryotes like the unicellular ciliate *Tetrahymena pyriformis* or the parasitic *Trypanosoma cruzi*, and even multicellular organisms including bivalves, vertebrates such as mammals and birds

(*Moschidis 1984*; *Ferguson, Allen & Snary, 1982*; *Keck et al., 2011*; *Smith, Snyder & Law, 1970*; *Tamari & Kametaka, 1972*).

Phosphonates are abundant especially in the marine environment (*Kolowith, Ingall & Benner, 2001*), and phosphonolipids specifically have previously been detected in marine invertebrates (*Imbs et al., 2021*). In bivalves, including species of *Mytilus*, *Crassostrea* and *Bathymodiolus*, phosphonolipids are highly abundant and have been used as tissue biomarkers (*Hori, Arakawa & Sugita, 1967*; *Kellermann et al., 2012*; *Sampugna et al., 1972*).

The phosphono-ceramides have been reported primarily in numerous species of marine invertebrates and seem to be common in this group of animals (*Hori, Arakawa & Sugita, 1967*; *Mukhamedova & Glushenkova, 2000*). The abundance of this lipid class underlines its importance - it was found to be one of the three major classes of phosphorous-containing lipids, next to PC and PE lipids, in a study of 32 mollusks (*Kostetsky & Velansky, 2009*). However, phosphonolipids cannot be assigned exclusively to one biological clade, all three mentioned lipid classes, PnE, PnC and PnE-Cer are also described in the insect *Cicada oni* (*Moschidis, 1987*).

Environmental and seasonal effects have previously been shown to influence the phosphonolipid content and composition in different species. A decrease in temperature led to an increase of phosphonolipids in cultured *Tetrahymena pyriformis* (*Hirobumi et al., 1976*). Oysters (*Crassostea virginica*) showed an increased relative abundance of phosphonolipids over phospholipids at the end of their reproductive cycle which could be reproduced by starving oysters in the laboratory (*Swift, 1977*). The selective conservation of phosphonolipids over phospholipids points towards an important function for the animal.

Despite their abundance in nature, the biological role of phosphonolipids is poorly understood, though a protective function has been suggested (*Kariotoglou & Mastronicolis, 1998*). Specifically, the incorporation of phosphonate moieties in cell–surface structures has been suggested as a protective feature (*Acker et al., 2022*; *White & Metcalf, 2007*). Phosphonates in general are resistant to abiotic hydrolysis by low pH and withstand boiling in concentrated acid (*Tamari & Kametaka, 1972*). Phosphonates can be potent inhibitors of enzymes while resisting hydrolysis because they are structurally similar to phosphate esters. Phosphonolipids in particular are resistant to hydrolysis by phospholipase enzymes (*Kafarski, 2019*) and therefore their catabolism requires phosphonate-specific enzymes. This chemical and biological stability could make phosphonolipids well suited as protective lipids against harmful environmental abiotic factors, such as heat and pH changes, and biotic factors, as most marine bacteria lack the enzymatic machinery to degrade phosphonates (*Villarreal-Chiu, Quinn & McGrath, 2012*).

As filter feeders, mussels can pump tens of liters of water through their gills and body cavity per day (*Riisgård, Egede & Barreiro Saavedra, 2011*). This feeding mechanism, unique to aquatic environments, exposes their epithelia to diverse microbes, including pathogens, and toxic metabolites (*Eggermont et al., 2017*). Phosphonolipids may improve the barrier function of mollusk tissues exposed to the environment. The accumulation of one such phosphonate lipid in epithelia exposed to the environment has recently been demonstrated in deep-sea mussels. High resolution spatial metabolomics revealed that one phosphonolipid is abundant in ciliated epithelial cells of the animal's gills, but absent from

non-ciliated neighboring cells that harbor bacterial symbionts (*Geier et al., 2020*). The stark contrast between the phosphonolipid's abundance in ciliated *vs.* colonized non-ciliated cells led us to the hypothesis that in mollusks phosphonolipids are enriched or even confined to ciliated epithelia, where they serve a protective role.

To investigate the hypothesis that phosphonolipids have a protective role, we analyzed lipid profiles of tissue samples from two species of marine bivalves, *Mytilus edulis* and *Crassostrea gigas.* Both *Mytilus* and *Crassostrea* species, better known as blue mussels and oysters, represent globally important fishery resources. We sampled gill and mantle tissue from both species, as well as the foot of *M. edulis.* These organs were chosen because they are outlined by a ciliated epithelial layer that is constantly in direct contact with the environment. Lipid extracts of these organs were screened by LC-MS and phosphonolipids were identified with MS$^2$. The spatial distribution of phosphonolipids in tissue sections was imaged and resolved to a pixel size of around 10 µm using high resolution atmospheric pressure scanning microprobe MALDI-MSI (AP-SMALDI-MSI) (*Römpp & Spengler, 2013*), which allowed us to locate lipids even within epithelial monolayers. Correlating spatial metabolomics data with optical microscopy on consecutive tissue sections subjected to histological staining allowed a clear co-localization of phosphonolipids with distinctly stained cell populations in gill, mantle and foot. MSI revealed that while some phosphonolipids are ubiquitously distributed, others are present only in the ciliated epithelial cells.

## MATERIALS & METHODS

### Animals

Live *Mytilus edulis* and *Crassostrea gigas* were purchased at a local store as live animals imported from France. Specimens were transported to the lab on ice to anesthetize the animals before the organs were dissected and separated for lipid extraction. Additional parts were cryo-embedded for mass spectrometry imaging and histology.

### Lipid extraction

Lipids were extracted from mussel tissues by a modified Bligh & Dyer protocol (*Bligh & Dyer, 1959*). Small pieces of mussel organs (∼50 mg) were submerged in methanol (8 µl * mg$^{-1}$ tissue) and subjected to mechanical lysis with silica beads (SiLiBeads Ceramic Beads Type ZY-S 1.1 - 1.2 mm diameter; Sigmund Lindner GmbH, Warmensteinach, Germany) in a bead-beating device (Fast-prep-24-5G, MP Biomedicals) for two 10 s bursts at 6.5 m * s $^{-1}$. The homogenized tissues were transferred into a three mL exetainer with chloroform (8 µl * mg$^{-1}$ tissue). The exetainers were vortexed for 15 s before HPLC-grade water (7.2 µl * mg$^{-1}$ tissue) was added and vortexed again for 30 s. Phase separation was allowed for 10 min. Cell debris was pelleted by a 10 min centrifugation step (4 °C, 2,500× g). The lipid fraction (organic solvent, lower phase) was transferred to a HPLC-MS vial *via* glass syringe. For analysis, 100 µL of a 1:10 dilution of the extract in acetonitrile was transferred to HPLC-MS vials (1.5-HRSV nine mm Screw Thread Vials; Thermo Fisher, Waltham, MA, USA). For each organ, triplicate samples were taken from three specimens

and a mixture of all samples from one species served as the quality control samples. All solvents were pre-chilled and the samples were kept on ice during the extraction procedure.

## LC-MS/MS

LC-MS/MS analysis was performed on a Vanquish Horizon UHPLC (Thermo Fisher Scientific) with Accucore C30 column (150 × 2.1 mm, 2.6 $\mu$m; Thermo Fisher Scientific, Waltham, MA, USA) at 40 °C connected to a Q Exactive Plus orbitrap mass analyzer with a HESI source (Thermo Fisher Scientific).

A solvent gradient of acetonitrile:water (60:40; vol./vol.) with 10 mM ammonium formate and 0.1% formic acid (buffer A) and 2-propanol:acetonitrile (90:10; vol./vol.) with 10 mM ammonium formate and 0.1% formic acid (buffer B) (*Breitkopf et al., 2017*) was used at a flow rate of 350 $\mu$l $\star$ min$^{-1}$. The gradient started at 0% buffer B and reached 97% B in 25 min, and was then followed by 7.5 min isocratic elution.

Per sample 10 $\mu$l extract were injected. MS measurements were acquired alternating between positive-ion and negative-ion mode in a range of *m/z* 150–1,500. The mass resolution was set to 70,000 for MS scans and 35,000 for MS/MS scans at *m/z* 200.

After each full MS scan dynamic data acquisition recorded MS/MS scans of the eight most abundant precursor ions with dynamic exclusion enabled for 30 s, followed by polarity switching. Fragments were generated by collision-induced dissociation (higher-energy C-trap dissociation) at an energy level of 30 eV. Raw data was analyzed with FreeStyle (Version 1.6.75.20; Thermo Fisher Scientific Inc., Waltham, MA, USA). LC-MS data are available at Metabolights (https://www.ebi.ac.uk/metabolights/) accession number MTBLS2960 (*Haug et al., 2020*).

Lipids were identified *via* MS/MS comparison and exact mass match using Lipidmaps as database query and match to theoretical sum formulas of phosphonolipids. Specifically, phosphono-ceramides are named by number in brackets for number of carbon atoms:number of double bonds in the fatty acid, -OH indicating a hydroxyl group on the sphingosine base.

## Mass spectrometry imaging sample preparation

Tissue samples were embedded in precooled 20 mg $\star$ ml$^{-1}$ carboxymethyl cellulose (MW ~700,000, Sigma Aldrich) and snap frozen in liquid nitrogen (*Kawamoto, 2003*). Tissue sections for MSI were prepared from embedded samples cut to a 10 $\mu$m thickness in a cryotome (Leica CM3050 S, −30 °C chamber temperature, −20 °C object holder temperature) and thaw-mounted on poly-L-lysine coated slides (Thermo Fisher). Slides were stored in a desiccator with silica beads (Carl Roth) under reduced pressure to avoid lipid oxidation before analysis. For all sections from *M. edulis* and *C. gigas*- gill and mantle tissue an ionization matrix composed of a mixture of 2,5-dimethoxy and 2-hydroxy-5-methoxybenzoic acid (Super-DHB, Sigma Aldrich) in acetone:water (60:40; vol./vol.) with 0.1% TFA was applied by the SMALDIPrep sprayer (TransMIT GmbH). Over 30 min 225 $\mu$l of a 30 mg $\star$ ml$^{-1}$ solution was deposited with nitrogen as carrier gas in a chamber containing a rotating sample slide. For one *M. edulis* foot section the ionization matrix was 2′,5′-Dihydroxyacetophenone (DHAP), applied as previously reported

(*Bien et al., 2021*) *via* a Sono-Tek SimCoat sprayer with ACCUMIST ultrasonic nebulizer (Sono-Tek Corporation, Milton, NY, USA). A 15 mg $^{*}$ ml$^{-1}$ solution of DHAP was sprayed with a flowrate of 50 $\mu$l$^{*}$min$^{-1}$ at a nitrogen pressure of 1 psi and ultrasonic frequency of 48 kHz. Per slide, 20 layers of matrix with a line distance of 1.8 mm were applied in a meandering pattern, alternating along the X- and Y- axis.

## Mass spectrometry imaging

Mass spectrometry imaging was done with an AP-SMALDI10 (TransMIT GmbH) ion source at atmospheric pressure coupled to an orbitrap mass spectrometer (Q Exactive HF; Thermo Fisher Scientific, Waltham, MA, USA). Laser focus was achieved by carefully adjusting the z-distance between sample and source until a minimal spot size was reached. All datasets were acquired at a resolution (pixel size) of 8–11 $\mu$m without oversampling. Spectral data was recorded in positive mode with a *m/z* range of either 350–1,500 or 300–1,200 and a mass resolution of 240,000 at *m/z* 200 (see Table S2).

## MSI data conversion and analysis

Raw data was converted to centroided .mzML format with MSConvert GUI (ProteoWizard, Version 3.0.9810) and subsequently to .imzML using the imzML Converter version 1.3.0 (*Race, Styles & Bunch, 2012*). Datasets where then imported to SCiLS Lab v2020b and total ion count (TIC) normalized ion maps were exported from SCiLS Lab for use in figures.

The imZML files of all acquired datasets have been uploaded to Metaspace2020 (http://www.metaspace2020.eu) (*Palmer et al., 2017*) and can be publicly browsed and downloaded. Colocalization analysis was performed based on the median-threshold cosine distance algorithm implemented in http://www.metaspace2020.eu (*Ovchinnikova et al., 2020*).

All raw data was uploaded to https://www.ebi.ac.uk/metabolights/. See Table S2 for datasets and settings for measurements.

## Histology

Consecutive tissue sections to the sections used for MSI, were prepared for histology and tissue-specific metabolite correlations. For histological analysis, tissue sections were stained with haematoxylin and eosin (H&E fast staining kit; Carl Roth, Karlsruhe, Germany). In brief, slides were submerged in solution 1 (modified haematoxylin solution), rinsed under de-ionized water for 10 s, submerged in 0.1% HCl for 10 s, blued under running de-ionized water for 6 min, submerged in solution 2 (modified eosin-g solution) for 30 s, and rinsed again for 30 s. Optical images were acquired on a slide scanning microscope (Olympus VS 120; Olympus Europa SE & Co. KG, Hamburg, Germany) in bright field with 20x magnification and exported as lossless PNG files.

## RESULTS

In total, we identified 20 different phosphonolipids with LC-MS/MS in tissues of *M. edulis* and *C. gigas* of which only two are represented in current lipid databases. We could further show the tissue distribution of the identified phosphonolipids with high resolution spatial

metabolomics. After comparing the spatial distribution of the phosphonolipids in the mollusks' tissues we found a subgroup of those lipids which co-localized specifically with epithelial tissue.

## Identification and structural diversity of phosphonolipids in marine mollusks

By analyzing total lipid extracts with LC-MS/MS we identified a diverse set of phosphonolipids from the class phosphono-ceramide (PnE-Cer) in the organs of two different bivalve species (see Fig. 1). Phosphonolipids' structural diversity is not reflected in lipid databases even though there is no obvious reason to expect their diversity to be lower than that of lipids with more commonly observed headgroups. Only two members of the PnE-Cer class are listed in LIPIDMAPS (accessed November 23rd 2021) (*Fahy et al., 2007*; *Sud et al., 2006*), one of the major current lipid databases. Thus, automated annotation was not suitable for identifying phosphonolipids in our samples. Instead, we manually screened the LC-MS/MS data of all parent ions in negative ionization mode for the characteristic fragment of the deprotonated phosphonate (*m/z* 124.0164)(*Facchini et al., 2016*) (see Figs. 1B, 1C). In positive ionization mode the neutral loss of the 2-aminoethylphosphonate (AEP) moiety *m/z* 125.0242 was used to confirm the identification of phosphonolipids (see Fig. 1D). Using this approach, we detected 20 PnE-Cer with 45 possible structural isomers in our five sample types (see Fig. 2, Table S1). The 20 phosphonolipids showed very different abundances across the species and organs, with signal intensities spanning three orders of magnitude. The most abundant phosphonolipids in the tissue extracts of *M. edulis* and *C. gigas* were PnE-Cer(32:1), PnE-Cer(34:2), PnE-Cer(35:3) and PnE-Cer(35:3)-OH. Among the less abundant phosphonolipids, we also detected PnE-Cer(34:1), the only phosphono sphingolipid besides PnE-Cer(32:1) included in LIPIDMAPS (entries: LM_ID LMSP04000002, LMSP04000001).

## Spatial distribution of phosphonolipids in different organs of *M. edulis and C. gigas*

Using spatial metabolomics with MSI we examined the distribution of the phosphonolipids identified with LC-MS/MS within organs of the two marine mollusks. We analyzed tissue sections of the gill and mantle of *M. edulis* and *C. gigas* as well as the foot of *M. edulis*. The 20 phosphonolipids showed variable spatial distributions even though they only differ by their fatty acid moieties (see Fig. 2). Some phosphonolipids were mostly confined to the epithelial layers of the organs, while others showed a homogenous distribution throughout the organs.

The most evident examples for a tissue-specific distribution, PnE-Cer(34:1) (*m/z* 667.5155, [M+Na]$^+$) in *M. edulis* and PnE-Cer(35:3-OH) (*m/z* 693.4948, [M+Na]$^+$) in *C. gigas*, were only present in epithelial tissue (see Fig. 3). In the mantle, a thin tissue layer lining the inside of the shell and covered by an epithelium, those two lipids were limited to the monolayer of epithelial cells at the rim of the organ (see Fig. 3). In the gills, the respiratory organ comprised almost entirely of epithelial cells with a ciliated surface, the epithelium-specific phosphonolipids were detected throughout the organ (Fig. S1).

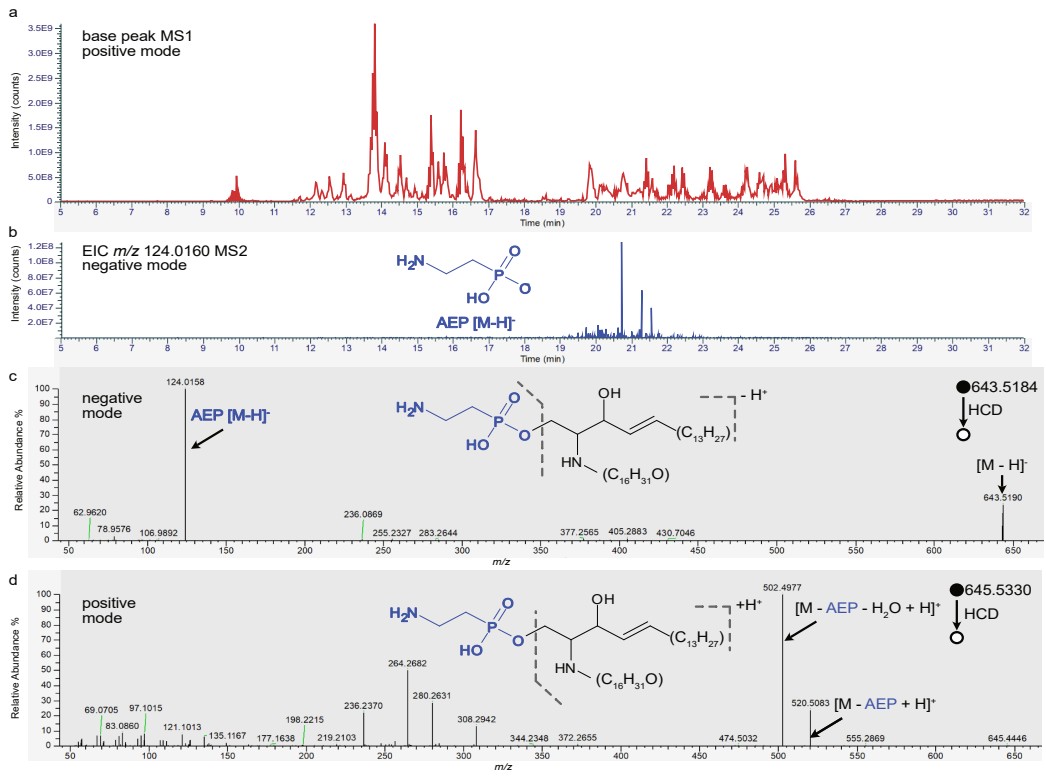

**Figure 1** **Mussel extracts contain a high number of phosphonolipids.** (A) LC-MS chromatogram (base peak) of *M. edulis* gill tissue extract, measured in positive mode. (B) Ion trace for the 2-aminoethylphosphonate (AEP) headgroup *m/z* $124.0160 \pm 5$ ppm in all acquired MS2 scans (negative mode). Putative identification of phosphonolipid PnE-Cer 34:1 (exact mass 644.525711) in (C) negative ionization mode (*m/z* 643.5184 [M-H]$^{-}$) and (D) positive ionization mode (*m/z* 645.5330 [M+H]$^{+}$). Indicative AEP occurs as neutral loss in positive ionization mode and as fragment ion in negative ionization mode (*m/z* 124.01).

PnE-Cer(35:3) (*m/z* 677.4999, [M+Na]$^{+}$) which is among the most abundant phosphonolipids in all analyzed tissues, showed a homogenous distribution throughout the organs. It is uniformly distributed in the mantle tissues of both *M. edulis* and *C. gigas* (see Fig. 3). A similar, homogenous distribution of PnE-Cer(35:3) was present in the gill tissues, where it was also detected in the basal membrane covered by endothelial cells. In those cells the epithelium-specific lipids PnE-Cer(34:1) and PnE-Cer(35:3)-OH were absent (see Fig. S1).

## Distribution of phosphonolipids in the foot of *Mytilus edulis*

The foot of *M. edulis* is the organ used for locomotion and chemical sensing, while in *C. gigas* the foot is regressed in adults (*Cannuel & Beninger, 2006*; *Lane & Nott, 1975*). Thus, a comparison of lipid distributions in foot tissues between the two bivalve species is not possible. We extended our study by testing a different matrix, dihydroxyacetophenone (DHAP) and a respective application protocol (*Bien et al., 2021*) which resulted in a higher number of metabolite annotations compared to sample preparation with DHB

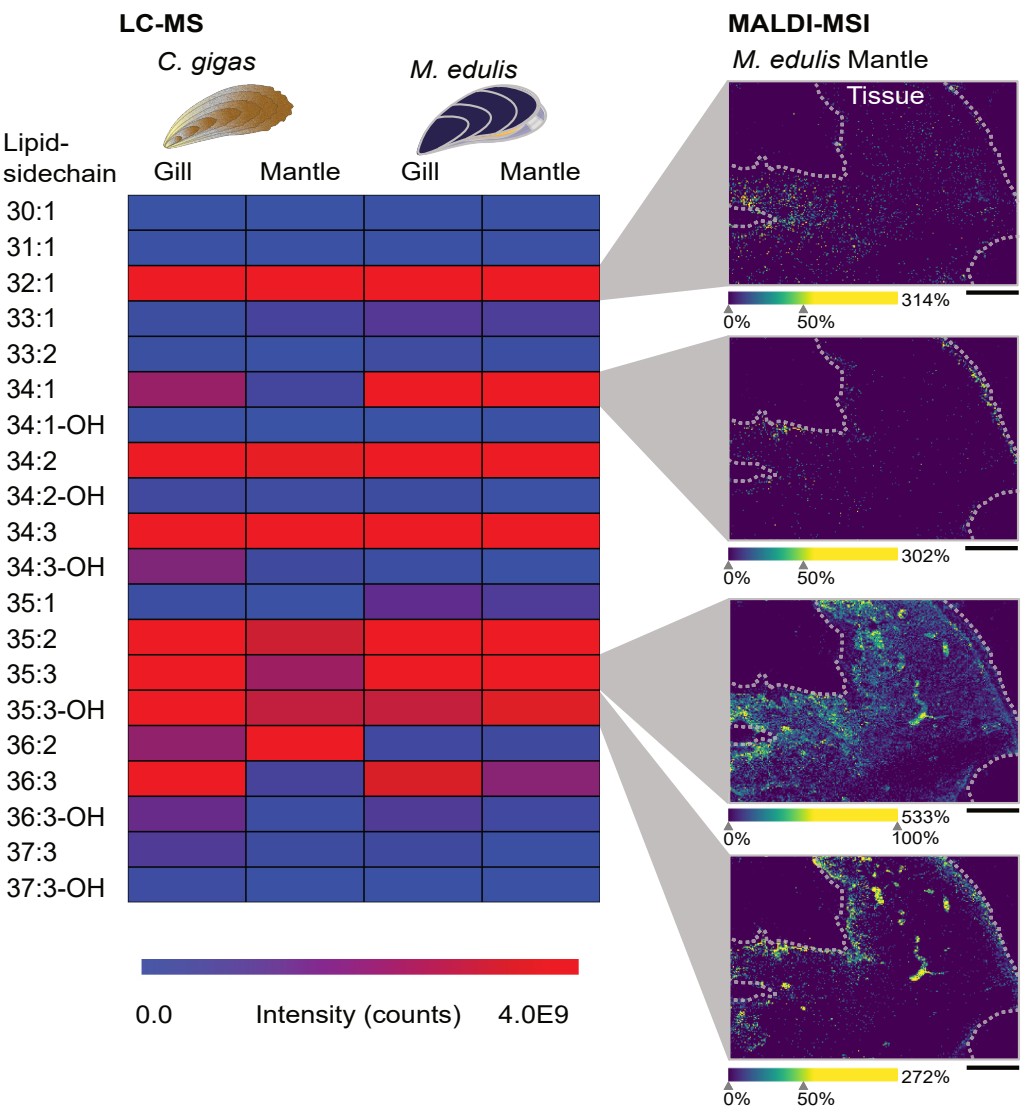

**Figure 2** **Relative abundance of identified phosphonolipids in analyzed mussels and selected metabolite distributions in *M. edulis* mantle tissue.** Heatmap table shows relative abundance of ion intensity counts from bulk LC-MS analysis, number of carbon atoms and double bonds in lipid backbone annotated as "Lipid-sidechain", isobaric isomers of a lipid are summarized and displayed as one value for the sum of all ions. For details of all [M-H]$^-$ ions, their sum formula and exact mass, see Table S1). Mass spectrometry imaging generated ion maps of selected abundant phosphonolipids in the mantle tissue of *M. edulis* (normalized to total ion count). Tissue outline indicated by dashed line. Scale bars: 400 μm.

(see Table S2). Phosphonolipids showed a tissue-specific distribution in the *M. edulis* foot sample (see Fig. 3), similar to the gill and mantle. In the foot we could again precisely localize the phosphonolipid PnE-Cer(34:1) (*m/z* 667.5155, [M+Na]$^+$) to the epithelial cells outlining the tissue (see Fig. 3J, 3K). The lipid was almost absent from other foot tissues, such as musculature and gland cells. The ubiquitous phosphonolipid PnE-Cer(35:3) and others were detected throughout the organ (see Figs. 3J, 3K).

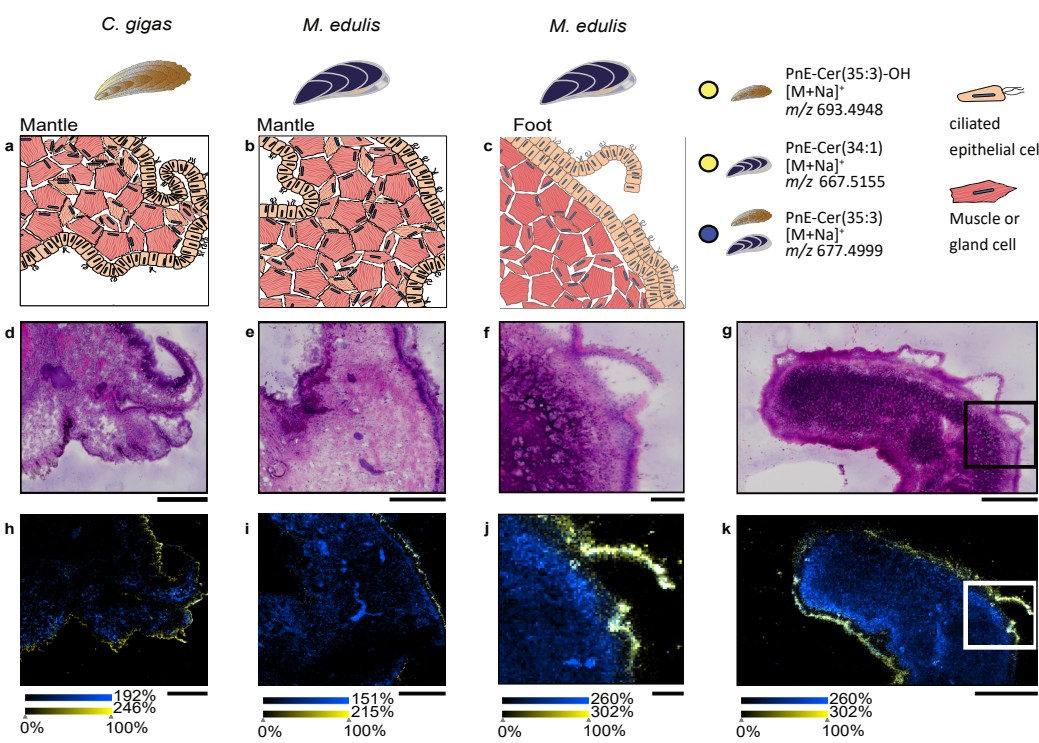

**Figure 3** **Tissue distributions of two different phosphonolipids.** (A–C) Schematic overviews of tissue sections, histology of *C. gigas* mantle (D), *M. edulis* mantle (E), *M. edulis* foot (F, G), tissues stained with H&E. Distribution of phosphonolipids in (H–K) both epithelia-specific phosphonolipids in yellow, ubiquitous phosphonolipid in blue. (H) yellow: PnE-Cer(35:3)-OH (*m/z* 693.4948 [M+Na]$^+$), (I–K) yellow: PnE-Cer(34:1) (*m/z* 667.5155 [M+Na]$^+$). (H–K) blue: Pne-Cer(35:3) (*m/z* 677.4999 [M+Na]$^+$). Scale bars: 500 μm (D, E, G, H, I, K); 100 μm (F, J).

As expected, other lipids such as phosphoserines (PS) are present throughout the organ, comparable to the phosphonolipids we classified as ubiquitous (see Fig. 4C).

To evaluate the distribution of epithelia-specific phosphonolipids in comparison to other lipids, we performed a colocalization analysis. We analyzed the *M. edulis* foot dataset as it provided the highest number of annotated lipids in the study (252 annotation @ FDR 10% in the LipidMaps database, see Table S2). A number of lipids were found with high colocalization values to the phosphonolipid that outlines the organ (PnE-Cer(34:1), *m/z* 667.5155, [M+Na]$^+$, see Fig. 4D). However, none of the 252 annotated metabolites showed the same fine scale distribution confined to the outer epithelial region of the tissue sections. This shows that the distribution of the epithelia-specific phosphonolipids is unique among lipids and points towards a specialized function.

## DISCUSSION

Despite their abundance in bacteria, protists and animals, especially in marine invertebrates, little is known about phosphonolipids biological functions. We discovered diverse spatial

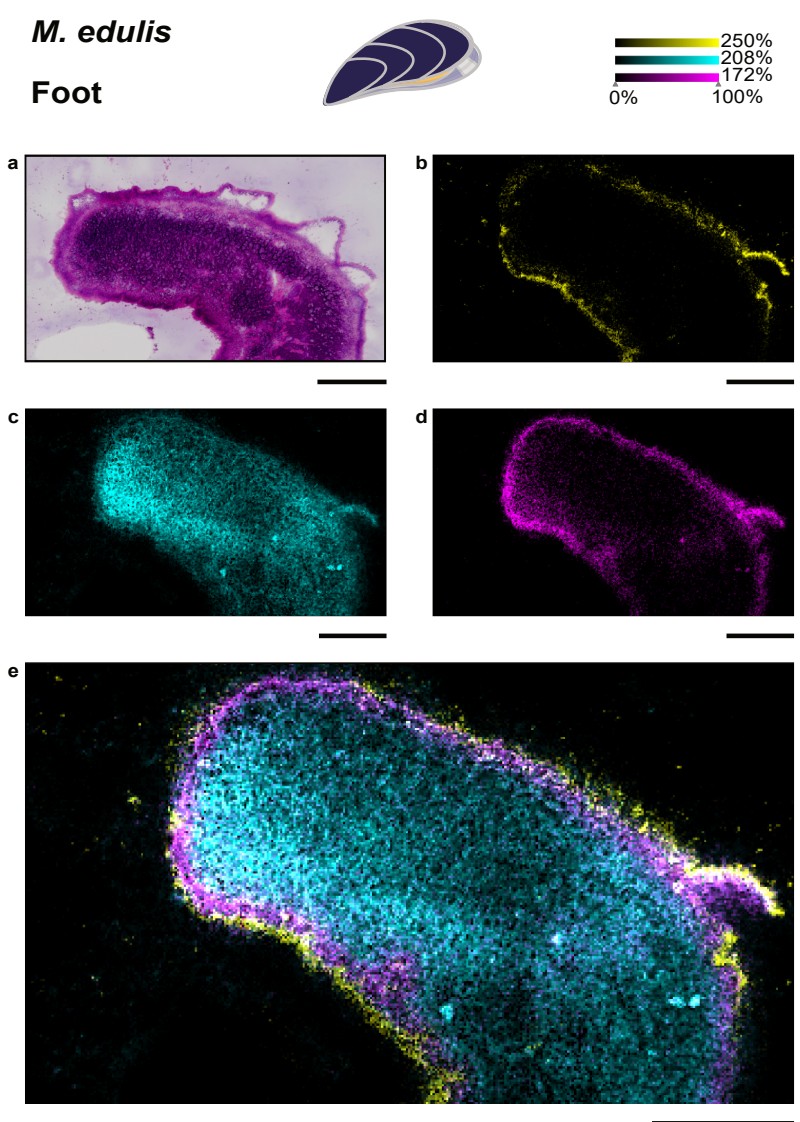

**Figure 4** **Specific lipid distributions in *M. edulis* foot.** (A) histology of *M. edulis* foot tissues stained with H&E, ion-maps showing the distribution of (B) epithelia-specific phosphonolipid (PnE-Cer(34:1) *m/z* 667.5155 [M+Na]$^+$) in yellow, (C) ubiquitous phospholipid (PS-(35:0) *m/z* 802.5359 [M+K]$^+$) in cyan (D) colocalized phospholipid (PC(36:4)/PS(O-38:4), m/z 820.5463 [M+Na]$^+$) in magenta, co-localization value = 0.41, highest colocaliztation value to epithelia-specific PnE-Cer(34:1), (E) overlay of all three lipids shown in (B), (C), (D). Scale bars: 500μm.

patterns of phosphonolipids of which some where highly tissue-specific. These distributions in the organs of marine mollusks possibly point towards specialized functions.

We found a high diversity of phosphonolipids in bivalve tissues and confirmed their identity with LC-MS/MS. Among phosphonolipids that were identified with high confidence we found 18 that are not covered in public databases, although these lipids have previously been reported in marine mollusks and other animals (*Facchini et al., 2016*; *Hori, Arakawa & Sugita, 1967*; *Kafarski, 2019*; *Kennedy & Thompson, 1970*; *Mukhamedova &*

*Glushenkova, 2000*). We envision that the addition of phosphonolipids to public databases including MS/MS spectra could substantially improve the annotation and identification of this class of so far little-studied lipids.

High-resolution spatial metabolomics revealed that phosphonolipids show unique distributions in bivalve tissues, allowing us to analyze the metabolic properties that underlie tissue functions, for example the epithelial barrier. While some lipid species are homogenously distributed others are spatially correlated with specific tissues within organs. Across different species of marine bivalves, we consistently found a subset of phosphonolipids matching the distribution of the ciliated epithelia outlining organs exposed to the environment. Across all analyzed organs, this subset of phosphonolipids was almost completely absent from other tissues. Similarly, In a deep-sea mussel species it was previously shown that the relative abundance of the phosphonolipid PnE-Cer(34:1) is five to ten times higher in the animals' gill compared to the foot (*Kellermann et al., 2012*). This can be explained by higher relative abundance of epithelial cells in the gill and partly confirms our hypothesis that phosphonolipids are confined to the tissues of the mollusks directly exposed to the environment.

## Could the enrichment of phosphonolipids provide stability and protection to cells in tissues exposed to the environment?

The most striking protective feature of bivalves is their shell. However, once the shell is opened and the mussel begins to filter feed and breathe, much of the animal's tissues are exposed directly to the environment. Those exposed and often mucous-covered soft tissues are a potential target site for microbial pathogens, which are constantly pumped along with water through the animals' mantle cavity. Lining the inner sides of the shell, the mantle of bivalves is the outermost organ of the animal covered by a ciliated epithelium. The gills, the respiratory organ of bivalves, are comprised almost entirely of epithelial cells and consist of many parallel filaments, arranged thusly to increase surface area and facilitate gas exchange. Extensive surface enlargement comes at a price; history shows that extensive borders are harder to defend against possible intruders. It is thus not surprising that both the gill and mantle have been shown to be sites of parasite infection in wild *Mytilus* mussels (*Mladineo et al., 2012*). In the tissue sections of *M. edulis* and *C. gigas* gills, the distribution of certain phosphonolipids follows the exposed cells. In these exposed cells a protective lipid with high chemical and biological stability, as exhibited by phosphonolipids, would be most effective.

Ciliation can act as a mechanical protection but also increases surface area and topography, and thereby possible infection sites. The cilia on gill and mantle epithelia of mussels generate a steady water current for respiration and transport food particles through ciliary movement (*Jones, Richards & Hutchinson, 1990*). Currently, MSI cannot spatially resolve if phosphonolipids are specific to just the ciliary membrane, or if they are also present in the epithelial cell membrane of ciliated cells. Without sub-micrometer resolution for MSI, additional experiments, such as LC-MS/MS on isolated cilia (*Mitchell, 2013*) would be needed. Notably, the analysis of cilia isolated through shearing forces revealed that phosphonolipids made up the majority of the ciliary lipids in a free-living protozoan

(*Smith, Snyder & Law, 1970*). Regardless of this finer differentiation, the abundance of some phosphonolipids on the outermost, exposed regions of organs could point towards a protective measure.

While the localization of a metabolite is no proof of its function, it applies for many biological examples, especially in the context of protective barriers against biotic and abiotic stressors in the environment. The spatial metabolite distribution, following a specific cell type or tissue, in many cases indicates the local function of a metabolite. Dotriacontanal, a wax lipid, is present only on the cuticle of maize plants where it protects from UV radiation and water loss (*Dueñas, Larson & Lee, 2019*). In the giant clam *Tridacna crocea*, UV-protective secondary mycosporines are localized in the outer layer of the epithelium (*Goto-Inoue et al., 2020*). A cocktail of antibiotics, produced by symbiotic bacteria, is found in high concentration only on the outer surface of the beewolf digger wasp cocoon it protects (*Kroiss et al., 2010*). The spatial distribution of these metabolites which shape tissues functionality, was revealed by MSI after their biological function was already known. We show that by applying MSI techniques, tissue-specific localization of metabolites complements our understanding of tissue functioning on a biochemical level. Teasing apart tissue chemistry is essential to translate and test our findings for potential applications in medicine or biotechnology. Understanding the role of phosphonolipids in defense against pathogens could open up a new line of research and identify a potential target for drug development to protect shellfish hatcheries.

## CONCLUSIONS

We were able to map the distribution of phosphonolipids in tissues of different environmentally and economically important mollusks species to reveal a possible role of phosphonolipids in those animals. For those phosphonolipids in the ciliated epithelia, a protective role is plausible, however, to prove this protective function, manipulative experiments are required. Future studies could investigate the fitness of pathogen-challenged mussels with and without phosphonolipids. Similarly, the infectivity of pathogenic *Vibrio* strains with a knock-out of phosphonate degradation genes should be tested against the wild type in a mussel infection system. In the future, correlative approaches with spatial metabolomics, such as spatial transcriptomics and proteomics, may resolve the biochemistry of phosphonolipid degradation during pathogen infection in mollusks.

We conclude that a functional annotation cannot be generalized for the entire class of lipids. It is evident from our study that phosphonolipids in marine mussels are chemically diverse and possibly have versatile functions based on their tissue distributions.

## ACKNOWLEDGEMENTS

We thank D. Jakob (MPI Bremen) for help with MSI measurements, M. Franke (MPI Bremen) for advice on mussel physiology and histology and O. Bourceau (MPI Bremen) for proof-reading the manuscript. We thank the K. Dreisewerd lab (Münster University)

for access to matrix application devices. We thank N. Dubilier (MPI Bremen) for access to lab resources and the constant support.

### Funding
This work was funded by the Max Planck Society, the MARUM Cluster of Excellence 'The Ocean Floor' (Deutsche Forschungsgemeinschaft (German Research Foundation) under Germany's Excellence Strategy—EXC-2077–39074603), and European Research Council Advanced Grant (BathyBiome, 340535). The funders had no role in study design, data collection and analysis, decision to publish, or preparation of the manuscript.

### Grant Disclosures
The following grant information was disclosed by the authors:
The Max Planck Society.
The MARUM Cluster of Excellence 'The Ocean Floor' (Deutsche Forschungsgemeinschaft (German Research Foundation) under Germany's Excellence Strategy—EXC-2077–39074603).
European Research Council Advanced Grant: BathyBiome, 340535.

### Competing Interests
The authors declare there are no competing interests.

### Author Contributions

- Patric Bourceau conceived and designed the experiments, performed the experiments, analyzed the data, performed the computation work, prepared figures and/or tables, authored or reviewed drafts of the article, and approved the final draft.
- Dolma Michellod performed the experiments, authored or reviewed drafts of the article, developed LC / MS workflow, and approved the final draft.
- Benedikt Geier conceived and designed the experiments, performed the experiments, authored or reviewed drafts of the article, and approved the final draft.
- Manuel Liebeke conceived and designed the experiments, authored or reviewed drafts of the article, and approved the final draft.

### Data Availability
All mass spectrometry imaging datasets are available in the open imZML format at www.metaspace2020.eu (Palmer, Phapale et al. 2017):
- MPIMM_136_QE_P_Mytilus_edulis_Mantle,
- MPIMM_137_QE_P_Mytilus_edulis_Gill,
- MPIMM_177_QE_P_Medulis,
- MPIMM_133_QE_P_Crassostrea_gigas_Mantle and
- MPIMM_134_QE_P_Crassostrea_gigas_Gill.
All LC-MS data and MSI data are available at MetaboLights: MTBLS2960.

## Supplemental Information

Supplemental information for this article can be found online at http://dx.doi.org/10.7717/peerj-achem.21#supplemental-information.

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
