# Peer review of "Spatial metabolomics shows contrasting phosphonolipid distributions in tissues of marine bivalves"

_PeerJ Analytical Chemistry, doi:10.7717/peerj-achem.21_

## Round 0.1 · original submission · Major Revisions

Besides the comments of the three reviewers, I want to add that you claim the importance of localization for biological function. However, no test, for example of antimicrobial activity, is presented in your paper.

·

Basic reporting

The manuscript entitled “Spatial metabolomics reveals functional diversity of phosphonolipids in marine bivalves” explains the detection and localization of lipids in two species of marine mollusks. Overall, the manuscript has an interesting approach combining MS imaging, histology, and LC-MS/MS techniques.
However, the manuscript explains not so well the methodology, the ionization technique is mentioned as MALDI but the instrument used in AP-SMALDI10 (TransMIT GmbH). The authors did not go in detail in the imaging part, the quality of the figures should be improved in order to reflect much better what is in the title of the manuscript "spatial metabolomics". The results reflect moderately what they hypothesized, phosphonolipids as protective molecules present on the outside mollusk tissues but should be explained and discussed in more detail. There are references poorly written, incomplete, and even nonpresent in the bibliography list. The figures have poor quality and contain errors.

Experimental design

The question and experimental design in principle follow what is expected in this type of analysis including a complementary approach using LC-MS/MS and a histological analysis.
In the subsection: MALDI-MSI sample preparation, it is not clear nor explained why the authors used two different approaches to apply the matrix, two different matrices and two instruments e.g. gill and mantle sections 2,5-dimethoxy and 2-hydroxy-5-methoxybenzoic acid (Super-DHB, Sigma Aldrich) was applied by the SMALDIPrep sprayer (TransMIT GmbH). For M. edulis foot 2′,5′-Dihydroxyacetophenone (DHAP) was applied via a Sono-Tek SimCoat sprayer with ACCUMIST ultrasonic nebulizer (Sono-Tek Corporation). The authors did not explain the reason, not in the experimental design not in the results or discussion sections.
In the subsection: MALDI-MSI measurements, Datasets were acquired with an AP-SMALDI10 (TransMIT GmbH) ion source, the ionization method needs clarification: MALDI or Atmospheric Pressure MALDI. The authors must clarify the resolution, the data conversion, and how the data were analyzed (open software, commercial one). imzML data from the foot of Mytilus edulis is missing in Metaspace2020.

Validity of the findings

The results section, including the figures, does not represent remarkable results. Figure 1, for example, is not a good figure and is poorly labeled (see Fig1a is written positive mode but in the caption, it is written negative mode), the figure does not contribute too much to be presented in the main text. Figure 2 uses too much space for the heat map and needs to be modified (see intensity bar). Figure 3 the intensity bar is illegible in all cases. In the caption "l" is erroneously added, but the last figure is marked as "k". More MS images of lipid distribution, comparison, and a discussion with possible protective function are expected in the results section.

Additional comments

The localization of molecules in tissues represents an extraordinary method to investigate and decipher the possible functions of molecules in organisms. I suggest that the authors re-evaluate the data already acquired, attention should be paid to the details, avoiding repetitive ideas. Improve the quality of the figures.

Reviewer 2 ·

Basic reporting

NO comment

Experimental design

No comment

Validity of the findings

The findings in this manuscript are novel and are of potential interest. The experiments are clearly presented, and the study question is certainly of great interest. The combined use of a metabolomic approach is a strength, much of the study is well done and the group is very capable of such laboratory analyses.

Additional comments

There are minor concerns that the authors need to address for the manuscript.
(1) It is not clear to the readers why authors choose gill, mantle tissue, foot for sampling?
(2) How do the authors rule out the possibility of the Phosphonolipid composition could be due to different environmental conditions or due to differences in seasonal modifications?
(3) Perhaps it was overlooked but could the authors please describe briefly about seasonal changes in phosphono lipid composition and also the interplay of environmental factors such as temperature and food quality.
(4) The authors have to mention if their study obtained ethical approval

·

Basic reporting

It is difficult read some of the labels on figures (Figure 1 and Figure 2). Please render images at high resolution and increase the font size.

Experimental design

No comment

Validity of the findings

1. It seems the authors have not thoroughly analyzed the relative abundance of Pne-Cer lipid subtypes in different organs or tissues of M. edulis and C.gigas. For eg: the localization of Pne-Cer 34:1 is not prominent in the mantle of C.gigas. Similarly, 34:2 is not abundant in the gill of C.gigas and 36:2 is not expressed in M.edulis. The authors need to evaluate these differential expressions within/in between the species and explain with respect to their phenotypes.

2. The authors stated that phosphonolipids only exist in invertebrates (Line 115-116). However, phosphonolipids are known to exist in variety of other species such as Tetrahymena pyriformis and single celled organisms (protozoa) - Trypanosoma cruzi. The phosphonolipids appear less abundantly in various bovine tissues and even in human aorta. Can authors comment on chemical composition. nomenclature differences, or side chain dissimilarities of these phosphonolipids in comparison to the lipids in mollusks species?

3. Although the authors precisely identify the distribution of PnE-Cer in the foot of M. edulis, they didn't correlate the functional significance. In order to do so, the authors must answer following questions:
• Why the phosphonolipid PnE-Cer (34:1) was absent in other foot cell types, such as musculature
and gland cells.
• What is the relative distribution of phosphonolipid Pne-Cer (35:3) and others in the foot compared
to gill and mantle of M.edulis?

Additional comments

1. The authors must differentiate the nomenclature of Pne-Cer phosphonolipids. For instance, Fig. 2 summarize the relative abundance of Pne-Cer lipids in mussels and mantle tissue. However, the sidechain nomenclature 34:2, 35:1, 36:2...etc are repeating throughout the plot.

2. Is there any chemical similarity between currently available therapeutics for shellfish hatcheries and phosphonolipids?

3. It is interesting to see that the phosphonolipids protect mucous covered soft tissues against microbial pathogen invasions. Do the phosphonolipids also protect against harmful UV rays and other environmental pollutants?

4. HMDB is wrongly abbreviated in the manuscript as HMBD. Please correct this minor thing.

---

## Round 0.2 · accepted · Accept

I would recommend rephrasing your part about the localization/function of the metabolites. I agree that a certain distribution indicates a biological function. Terming this 'the Bauhaus principle of design “form follows function” ' seems to me 'ziemlich weit hergeholt', since Bauhaus is a concept of architecture. I would better write "suggests/indicates" a biological function.

·

Basic reporting

no comment

Experimental design

no comment

Validity of the findings

no comment

Additional comments

The authors addressed the reviewers' questions and comments.
The authors clarified the experimental part, improved the figures and typing errors, and added important information in the introduction and discussion.
The manuscript improved considerably.

·

Basic reporting

no comment

Experimental design

no comment

Validity of the findings

no comment

Additional comments

The authors have addressed all my concerns and thoroughly revised the manuscript. So, I don't have any additional comments.